# Bmp4 in Zebrafish Enhances Antiviral Innate Immunity through p38 MAPK (Mitogen-Activated Protein Kinases) Pathway

**DOI:** 10.3390/ijms241914444

**Published:** 2023-09-22

**Authors:** Lihui Chen, Shenjie Zhong, Yajun Wang, Xinyuan Wang, Zhenhui Liu, Guobin Hu

**Affiliations:** College of Marine Life Science, Institute of Evolution & Marine Biodiversity, Ocean University of China, Qingdao 266003, China; h19862193554@163.com (L.C.); zhongshenjie@ouc.edu.cn (S.Z.); wangyajun419@163.com (Y.W.); WangXinyuan991208@163.com (X.W.)

**Keywords:** zebrafish, BMP4, antiviral innate immunity, IFN, p38 MAPK pathway

## Abstract

Bone morphogenetic proteins (BMPs) are a group of structurally and functionally related signaling molecules that comprise a subfamily, belonging to the TGF-β superfamily. Most BMPs play roles in the regulation of embryonic development, stem cell differentiation, tumor growth and some cardiovascular and cerebrovascular diseases. Although evidence is emerging for the antiviral immunity of a few BMPs, more BMPs are needed to determine whether this function is universal. Here, we identified the zebrafish *bmp4* ortholog, whose expression is up-regulated through challenge with grass carp reovirus (GCRV) or its mimic poly(I:C). The overexpression of *bmp4* in epithelioma papulosum cyprini (EPC) cells significantly decreased the viral titer of GCRV-infected cells. Moreover, compared to wild-type zebrafish, viral load and mortality were significantly increased in both larvae and adults of *bmp4*^−/−^ mutant zebrafish infected with GCRV virus. We further demonstrated that Bmp4 promotes the phosphorylation of Tbk1 and Irf3 through the p38 MAPK pathway, thereby inducing the production of type I IFNs in response to virus infection. These data suggest that Bmp4 plays an important role in the host defense against virus infection. Our study expands the understanding of BMP protein functions and opens up new targets for the control of viral infection.

## 1. Introduction

Bone morphogenetic protein (BMP) is a member of the transforming growth factor β (TGF-β) superfamily. It was originally recognized as a factor in the body that can induce bone and cartilage formation, also known as “osteoblast”. BMPs and their corresponding receptors have been found in almost all organs during animal growth and development. Therefore, the function of BMPs goes far beyond mere bone induction. BMPs not only have obvious osteogenic effects, but also participate in the genesis and development of tumors, can inhibit cell growth and proliferation and can promote cell differentiation and apoptosis [1,2,3,4,5].

Human BMP4, as a member of the BMP family, has the same spatial structure as BMPs. The BMP4 preproprotein is composed of about 400 amino acids and consists of three parts: N-terminal signaling peptide, preprotein folding region and C-terminal maturation peptide. The carboxy-terminally mature BMP4 protein can be cleaved from the preproprotein to become a highly conserved BMP4 molecule consisting of 116 amino acids. Its C-terminal contains seven cysteine residues, six of which form three intramolecular disulfide bonds, called cysteine junctions, and the seventh cysteine can be glycosylated to form a covalent disulfide bond for dimerization with another monomer, thus forming a bioactive signaling molecule [6]. BMP4 plays important roles in promoting bone tissue regeneration and repair. In addition, BMP4 is also closely related to inducing embryonic differentiation, guiding neural stem cell differentiation, regulating tumor growth and invasion, and some cardiovascular and cerebrovascular diseases [7,8,9,10].

In recent years, the unappreciated roles of BMPs in the regulation of antiviral immunity have been demonstrated [11,12,13,14,15,16]. BMP6 can enhance antiviral responses and suppress the growth of Hepatitis B virus (HBV) in cell culture [17]. BMP2 may have a role in inducing virus-induced and interferon-associated responses [18]. We also found that Bmp8a plays a regulatory role in the antiviral immune response [19]. However, whether BMP4 has a function in the regulation of immune responses is unknown. Zebrafish has been developed as a simple vertebrate system for large-scale genetic and evolution analysis [20,21]. In the present study, zebrafish was used as a model to answer this question. We demonstrated that Bmp4 plays a role in the host defense against virus infection.

## 2. Results

### 2.1. The Identification and Evolution of Zebrafish Bmp4

The gene of zebrafish *bmp4* was acquired from the Ensembl database (Ensembl: ENSDARG00000019995, http://www.ensembl.org, accessed on 10 May 2021). This gene has one transcript encoding a protein of 400 amino acids. By searching in the National Center for Biotechnology Information (NCBI) genome database (https://www.ncbi.nlm.nih.gov/, accessed on 10 May 2021), we found that orthologs of this gene were present in mammals, birds, amphibians, teleosts and cartilaginous fish (see Figure 1 legend for the gene source information). The phylogenetic tree based on the amino acid sequences of these BMP4 proteins showed that the BMP4 molecules of mammals, birds and amphibians may have evolved from fish BMP4 (Figure 1A). The comparison of the amino acid sequences, exon–intron organizations and three-dimensional (3D) structures among them showed that they are very similar (Figure 1B–D and Figure 2A). Apparently, the sequences of their maturation peptides have the highest identity (Figure 1B). They all contain seven cysteine residues in their C-terminal (Figure 1B). In addition, the *bmp4* loci in *D. rerio* are adjacent to the genes *ddhd1a* and *fermt2*, which resemble the *BMP4* regions of other species (Figure 2B). Both the order and direction of these genes are similar among these species, indicating the synteny-encompassing *BMP4* region during vertebrate evolution.

### 2.2. The Bmp4 Expression Was Increased after Virus or Poly(I:C) Challenge

We first detected the distribution of *bmp4* mRNA in tissues of adult zebrafish through quantitative reverse-transcription PCR (qRT-PCR) technology. It was found that zebrafish *bmp4* mRNA was expressed in all the detected tissues (Figure 3A), implying multiple functions of this gene. Obviously, *bmp4* had the higher expression in the tissues of the ovary, testis, skin, gill and eye (Figure 3A). Then, the expression of *bmp4* upon virus infection was measured in the immune-related tissues including the gill, liver, spleen, kidney and intestine in zebrafish. After 24 h of stimulation with GCRV (a dsRNA virus), the expression of *bmp4* in the tissues of the gill, liver and intestine was significantly increased (Figure 3B). The in vitro effect of GCRV or poly(I:C) challenge on bmp4 expression was also detected in zebrafish liver (ZFL) cells. GCRV or poly(I:C) significantly elevated the expression of bmp4 in ZFL cells (Figure 3C,D). These suggest that the expression of bmp4 is induced by infection with virus or its mimic poly(I:C).

### 2.3. Antiviral Function of Bmp4 Both In Vitro and In Vivo

We then explored the role of zebrafish Bmp4 in antiviral immunity. The wild-type EPC cells and EPC cells over-expressing bmp4 were infected with GCRV virus. We found that, after 24 h of GCRV stimulation, a significant decrease in viral titers was observed in the EPC cells overexpressing *bmp4* (Figure 4). In addition, the CRISPR-Cas9 technology was used to generate bmp4-deficient (*bmp4*^−/−^) zebrafish, in which 1 nucleotide was replaced by 11 nucleotides in exon 3 of bmp4, leading to early termination of translation to the 118th amino acid (Figure 5A). The bmp4 knockout zebrafish (*bmp4*^−/−^) can survive, reproduce and breed normally under standard laboratory conditions. To study the function of Bmp4 in antiviral immunity in vivo, the wild-type and *bmp4*^−/−^ zebrafish larvae were infected by adding GCRV in the embryo culture medium. Compared with wild-type zebrafish, *bmp4*^−/−^ zebrafish larvae exhibited a significantly reduced survival rate upon GCRV infection, and most of the fish died 36 h post infection, while only few wild-type zebrafish died during the period (Figure 5B). Moreover, the adult zebrafish were infected with the virus of GCRV (Figure 5C). After infection with GCRV, the survival rate of *bmp4*^−/−^ zebrafish was significantly reduced (Figure 5D). Also, the levels of virus RNA of GCRV in the liver and kidney of bmp4^−/−^ adult zebrafish were significantly increased compared to those of wild-type zebrafish (Figure 5E,F). These data suggest that Bmp4 plays a role in the host defense against virus infection.

### 2.4. Bmp4 Increases the Expression of Antiviral Genes

The bmp4 was over-expressed in EPC cells, and then qRT-PCR was used to detect the expressions of type I interferon (IFN) genes and the antiviral protein gene mx. It was found that EPC cells with over-expressed bmp4 showed a markedly higher expression of *ifn* and mx than that of control cells, infected with or without GCRV (Figure 6A–D). Moreover, when both the wild-type and *bmp4*^−/−^ zebrafish larvae or adult zebrafish were challenged with GCRV, the expression of both ifn*φ*1 and mxa was all significantly down-regulated in the *bmp4*^−/−^ zebrafish larvae or the liver and spleen of bmp4^−/−^ adult zebrafish, compared with those of wild-type fish (Figure 6E–H). All these data indicate that BMP4 promotes the expression of antiviral genes.

### 2.5. Bmp4 Activates Antiviral Signaling via p38 MAPK Pathway

To explore the molecular mechanism of Bmp4 in antiviral immune regulation, we injected GCRV intravenously into *bmp4*^−/−^ mutant and wild-type zebrafish, and then performed transcriptomic analysis of their livers. Kyoto Encyclopedia of Genes and Genomes (KEGG) analyses revealed that the differential expressed genes (DEGs) were enriched in the MAPK signaling pathway, cytokine–cytokine receptor interaction and focal adhesion that may be involved in the immune process (Figure 7). It has been known that the TBK1-IRF3/7-IFN signal plays a significant role in antiviral innate immune responses, so we then evaluated the activation of Tbk1 and Irf3 using immunoblot assays in both *bmp4*-overexpressing and wild-type EPC cells, infected with or without GCRV. It was found that overexpressed *bmp4* significantly increased phosphorylation levels of Tbk1 and Irf3 (Figure 8A,B). This was further supported by the observations in vivo that the expression of *tbk1* and *irf3* was significantly downregulated in *bmp4*^−/−^ mutant than that in wild-type zebrafish larvae (Figure 8C,D). This suggests that BMP4 may activate Tbk1-Irf3 antiviral signaling.

To further examine the signal pathway by which Bmp8a stimulates ifn expression, the effects of p38 MAPK inhibitor SB203580, JNK inhibitor SP600125, MEK1/2 inhibitor U0126, SMAD1/5/8 inhibitor DMH1 and smad2/3 inhibitor TP0427736 HCl on the expression of ifn were tested in bmp4-overexpressing EPC cells. We found that p38 MAPK inhibitor SB203580 remarkably reduced the expression of ifn in EPC cells (Figure 8E). We then evaluated the activation of p38 MAPK through immunoblot assays in both *bmp4*-overexpressing and wild-type EPC cells, infected with or without GCRV. It was found that phosphorylation levels of p38 MAPK were significantly increased in the cells with overexpressed *bmp4* (Figure 8F). Collectively, the data indicate that Bmp4 induces the expression of ifn through the p38 MAPK pathway.

## 3. Discussion

BMP4 is an extracellular polypeptide signaling molecule of the BMP family. It can be found in jawed vertebrates including mammals, birds, amphibians, teleosts and cartilaginous fish (Figure 1A). However, we could not identify the *BMP4* orthologs in jawless vertebrate lamprey. When the zebrafish Bmp4 sequence was used as a bait to search the orthologs in the sea lamprey *Petromyzon marinus* genome, Bmp2 was shown to be the closest to Bmp4. Thus, it is highly possible that *bmp4* genes were first present in cartilaginous fish and originated from *bmp2* during evolution. 

In *H. sapiens*, *BMP4* is highly expressed in the placenta, urinary bladder, prostate, ovary, colon and stomach (human ENCODE transcriptome data in NCBI), while *Bmp4* mRNA in *M. musculus* is mainly present in the bladder, lung, kidney and intestine including the small intestine, colon and large intestine (mouse ENCODE transcriptome data in NCBI). In this paper, we showed that the zebrafish *bmp4* is mainly expressed in the tissues of the ovary, testis, skin, gill and eye (Figure 3A). The expression patterns of *BMP4* appear to differ across species, but it is also clear that BMP4 is expressed in almost all tissues of different species, which supports the multifunctional growth factor of BMP4.

As a model animal, zebrafish is becoming a powerful tool in the research of lots of fields [20,21]. In this study, we demonstrated that the knockout of *bmp4* in zebrafish results in the increase in the viral load and mortality after infection with GCRV virus in both larvae and adults (Figure 5), indicating its antiviral function. It is well known that IFNs play important roles in inhibiting virus replication by inducing the expression of antiviral proteins via the Janus kinase signal transducer and activator of transcription (JAK-STAT) pathway [22]. In zebrafish, it has revealed that *ifnφ1* and *ifnφ3* were identified to have antiviral responses [23], while only one type I IFN was shown to be involved in the antiviral role in carp and EPC cell [24,25]. In this paper, we demonstrated that Bmp4 promotes the expression of antiviral genes *ifn* and mx in EPC cells, while the expression of *ifnφ1* and *mxa* was decreased in *bmp4*-deficient zebrafish (Figure 6). These data suggest that zebrafish *bmp4* plays roles in the regulation of antiviral innate immune responses. Very recently, we have also demonstrated that Bmp8a is a positive regulator in antiviral immune responses [19]. In addition, BMP6 was shown to block HCV (Hepatitis C virus) replication [17]. Thus, our finding here expands the understanding of the roles of BMP molecules in antiviral immune responses. It will be interesting to uncover whether other members of the BMP family have the function in antiviral immune responses in the future. In fact, the immunoregulatory roles of BMP members have also been reported [26]. For example, the BMP pathway is one of the most significant contributors to the cell–cell interaction network among mouse immune cells in the reconstructed communication network [27]. A deficiency of BMP2, 4 or 7 results in decreased thymus size [16,17,28,29,30,31]. Moreover, BMPs, TGFβ and activins constitute the TGF-β superfamily. TGFβ can act in the regulation of inflammatory effects [32]. The antiviral activity of activins have also been shown [17]. Therefore, the immunoregulatory function of more members of the TGF-β superfamily would be expected. 

Viral infection can be detected by pattern recognition receptors (PRRs) in host cells that trigger the activation of IRF3 and IRF7, thereby inducing the production of type I IFN and IFN-stimulated genes (ISGs) [33,34,35,36]. Here, we demonstrated that Bmp4 increases the expression of antiviral genes *ifn* and *mxa* in vitro and in vivo, indicating a role of BMP4 in the antiviral signals. BMPs are known to perform different functions through either the Smad-dependent pathways (SMAD1/5/8 and SMAD2/3 pathways) or the Smad-independent pathways (ERK, JNK and p38 MAPK pathways) [37,38,39]. Our study reveals that the p38 MAPK pathway may be the main pathway for BMP4 to exert antiviral immunity. p38 MAPKs are conserved serine-threonine protein kinases that can be activated by a variety of extracellular inflammatory factors (for example, TNF-α and IL-1), bacterial lipopolysaccharide (LPS) and chemokines [40]. Numerous studies have shown that p38 MAPK activity is essential for normal immune and inflammatory responses [41,42]. Our finding expands the understanding of the role of p38 MAPK in the regulation of the immune response. 

A previous study has shown that MAPK signaling increases the phosphorylation of TBK1 and IRF3 upon viral infection [43]. Here, we also revealed that Bmp4 significantly increased phosphorylation levels of Tbk1 and Irf3. Taking all the data into consideration, we suggest that Bmp4 activates Tbk1-Irf3-Ifn antiviral signaling via the p38 MAPK pathway, eventually resulting in increased synthesis of type I IFN (Figure 8G). This is similar to the regulatory pathway of BMP8A in antiviral immunity [19]. However, additional pathways independent of p38 MAPK-Tbk1-Irf3-IFN are not excluded. In fact, BMP6 not only enhances transcription and the antiviral response to IFN, but also inhibits HCV replication independently of IFN [17]. In addition, which types of receptors are responsible for binding to BMP4, thereby triggering an antiviral immune response, will require further investigation. BMP type I receptor Alk6a has been shown to participate in the antiviral immunity [19], so this receptor may be one of the candidates. 

The expression of zebrafish *bmp4* is significantly upregulated after challenge with GCRV and poly(I:C). We have shown that STAT1 can directly bind to the IFN-γ activation sites (GAS) in the *bmp8a* promoter, suggesting that *bmp8a* can be activated by viruses via the Ifn-γ-Jak-Stat1 pathway [19,43,44,45]. Multiple GAS sites in the *bmp4* putative promoter region were also found when searching for the transcription factor STAT1 binding sites on the web (http://jaspar.genereg.net/, accessed on 22 June 2023). It is highly likely that the transcripts of *bmp4* can be activated by Stat1. Thus, it is possible that *bmp4* can also be activated by viruses via the Ifn-γ-Jak-Stat1 pathway.

## 4. Materials and Methods

### 4.1. Cell Culture

We bought the epithelioma papulosum cyprini cells (EPC) and zebrafish liver cells (ZFL) from the China Zebrafish Resource Center (CZRC). EPC cells were cultured in MEM media supplemented with 10% fetal bovine serum (FBS) (Gibco), 100 U/mL of penicillin and 100 μg/mL of streptomycin. ZFL cells were maintained in DMEM/F-12 media supplemented with 10% FBS, 100 U/mL of penicillin and 100 μg/mL of streptomycin. They were grown at 28 °C in an incubator supplied with 5% of CO_2_.

### 4.2. Viruses and Poly(I:C)

Grass carp reovirus (GCRV) was kindly provided by professor Yibing Zhang at the Institute of Hydrobiology, Chinese Academy of Sciences. The GCRV is a dsRNA virus. The 50% tissue culture-infective dose (TCID_50_) assay was performed to determine the viral titer in EPC cells [46]. Poly(I:C) is a synthetic double-stranded RNA (dsRNA) analog that is a molecular pattern associated with viral infection. ZFL cells were transfected with poly(I:C). After 1 h, the expression of *bmp4* was detected using qRT-PCR technology.

### 4.3. Sequence Comparison and Phylogenetic Analysis of BMP4 Proteins

Multiple BMP4 protein sequences were aligned using the Clustal W method in the DNASTAR software (version 7.1) [47]. The exon–intron information of *BMP4* genes was obtained from the Ensembl database (http://www.ensembl.org, accessed on 1 July 2023). The Sequence Viewer (http://www.ncbi.nlm.nih.gov/projects/sviewer/, accessed on 1 July 2023) and Ensembl Genome Browser were used to retrieve the synteny data among BMP4 orthologs in different species. The three-dimensional structure of BMP4 proteins was predicted using the SWISS-MODEL program of the Expert Protein Analysis System online (http://www.expasy.org/, accessed on 1 July 2023) [48]. MEGA (version 11) software was used to construct the phylogenetic tree with the neighbor-joining method. Sea lamprey Bmp2 was selected as an outgroup for rooting. Bootstrap was performed 1000 times to estimate the confidence of each node.

### 4.4. Generation of Bmp4 Mutant Zebrafish

The zebrafish *Danio rerio bmp4* knockout mutant lines (*bmp4*^−/−^) were established from the zebrafish AB line using CRISPR/Cas9 technology [49]. The *bmp4* target in this study was 5′-agtcgagccaacaccgtgag-3′ in the third exon. Mixtures of Cas9 protein and gRNA were microinjected into 1-cell-stage zebrafish embryos. Mutant sites were verified by comparison to the WT zebrafish sequences. Heterozygous F1 mutants of the same mutation were crossed to obtain homozygous germline mutants. The primers used in this study are listed in Table 1.

### 4.5. Plasmid Construction

For the eukaryotic expression, the coding sequence (CDS) of zebrafish *bmp4* was amplified through PCR and cloned into pcDNA3.1(+) vector. The primer sequences are listed in Table 1.

### 4.6. Transcriptome Sequencing and Identification of DEGs

Adult zebrafish were randomly divided into wild-type group and *bmp4*^−/−^ group. Each fish was intraperitoneally injected (i.p.) with 40 μL of GCRV (2.5 × 10^7^ TCID_50_/mL). After 36 h, liver tissue from three fish in each group was collected together as a separate sample for total RNA extraction. Transcriptome sequencing was performed using BioMarker (Beijing, China). RPKM (reads per kb per million reads) method was used to calculate and normalize gene abundance. DEGs (differentially expressed genes) between groups were determined using the edgeR package (http://www.r-project.org/, accessed on 28 July 2023). Fold change ≥2 or ≤0.5, together with false discovery rate (FDR) <0.05, was considered statistically significant. Then, the enrichment analysis of KEGG (Kyoto Encyclopedia of Genes and Genomes) pathway was performed.

### 4.7. Viral Infection In Vitro

EPC and ZFL cells were infected with GCRV (10^6^ TCID_50_ per mL) in the medium without FBS (Gibco), and after 1 h, they continued to be cultured in 2% culture medium. The virus was collected until the cytopathic effect reached around 70% for subsequent experiments. 

### 4.8. Viral Infection In Vivo

Zebrafish larvae were cultured in embryo culture medium (CaCl_2_ 0.4 g, KCl 0.3 g, MgSO_4_ 0.79 g, NaCl 10 g in 10 L of deionized water). GCRV was added to the embryo culture medium (final concentration 6 × 10^6^ TCID_50_/mL) to infect WT and *bmp4*^−/−^ zebrafish larvae (4 dpf). In addition, WT and *bmp4*^−/−^ adult zebrafish were infected with 40 μL of GCRV (2.5 × 10^7^TCID_50_/mL) per fish through intraperitoneal injection. Total RNA was extracted from adult zebrafish and the expression of viral RNA in liver and spleen was detected through qRT-PCR. The survival of zebrafish infected with GCRV was monitored at different time points as indicated. 

### 4.9. Crystal Violet Staining

EPC cells were transfected with zebrafish *bmp4* and empty vector plasmid. After transfection for 24 h, the cells were challenged with GCRV (10^6^ TCID_50_/mL). After 48 h of challenge, cells were fixed with 4% formaldehyde for 2 h, followed by staining with 0.5% crystal violet for 2 h. After flushing with water, the experimental results were observed and photographed.

### 4.10. Inhibitor Treatment and Analysis

The inhibitors used in this study were SMAD1/5/8 inhibitor DMH1 (Selleck, Houston, TX, USA, #S7146), smad2/3 inhibitor TP0427736 HCl (Selleck, Houston, TX, USA, #S8700), p38 MAPK inhibitor SB203580 (Beyotime, Shanghai, China, #S1863), JNK inhibitor SP600125 (Beyotime, Shanghai, China, #S1876) and ERK1/2 inhibitor U0126 (Cell Signaling Technology, Danvers, MA, USA). They were dissolved in dimethyl sulfoxide (DMSO). EPC cells were treated with inhibitors (SP600125, U0126 and DMH1) in a concentration of 10 μM for 24 h, and SB203580 and TP0427736 HCl were used in 5 μM for 24 h. Total RNA was extracted from the cells for qRT-PCR analysis.

### 4.11. Quantitative Real-Time PCR

Total RNA from zebrafish or EPC cells was extracted using the Total RNA Kit I (Omega, Guangzhou, China). RNA was treated with DNase. Then, PrimeScript™ RT reagent Kit with gDNA Eraser (TaKaRa, Dalian, China, #RP047A) was used to synthesize cDNAs. At the same time, samples without reverse transcriptase were added as controls. To detect the gene expression, the cDNA was amplified using ChamQ SYBR Color qPCR Master Mix (Vazyme, Nanjing, China, #Q431-02) using ABI 7500 Fast Real-Time PCR System (Applied Biosystems, Foster City, CA, USA). The expression of zebrafish actb1 or EPC actin was used as the internal reference. The relative expression was calculated using the 2^−ΔΔCt^ method. All qRT-PCR experiments were triplicate and repeated three times. Primers used in this study are listed in Table 1.

### 4.12. Western Blot and Antibodies

Western blot was performed according to a previous description [19]. In brief, whole-cell lysates were separated by 12.5% SDS-PAGE gels, electrotransferred to PVDF membrane, blocked with 5% fat-free milk solution for 2 h at room temperature, and incubated with the primary antibody at 4 °C overnight. Non-specific binding and excess antibodies were washed using 1× PBST, incubated with the second antibody at room temperature for 1 h and detected through enhanced chemiluminescence (ECL). Antibodies’ information is listed in Table 2.

### 4.13. Statistical Analysis

Graphpad Prism 9.0.0 software was used to perform all statistical analyses. Data are expressed as mean ± SD. Statistical significance between two groups was analyzed through two-tailed Student’s *t*-test. For the comparison of multiple groups, one-way ANOVA was used for statistical analysis, and then Games–Howell post hoc test was performed. Survival was analyzed using Kaplan–Meier method and evaluated using log-rank (Mantel–Cox) test. * *p* < 0.05, ** *p* < 0.01, *** *p* < 0.001; ns, not significant, *p* > 0.05.

## 5. Conclusions

Zebrafish Bmp4 is demonstrated to have roles in antiviral innate immune responses in this paper. The transcripts of bmp4 are activated upon viral infection. The *bmp4*^−/−^ mutant zebrafish significantly increases the viral load and mortality after infection with GCRV virus in both larvae and adults. Furthermore, Bmp4 promotes the phosphorylation level of Tbk1 through the p38 MAPK pathway and increases the production of type I IFN. Our study expands the understanding of the function and regulation of BMPs in antiviral immune responses and may provide new targets to confront viral infection.

## Figures and Tables

**Figure 1 ijms-24-14444-f001:**
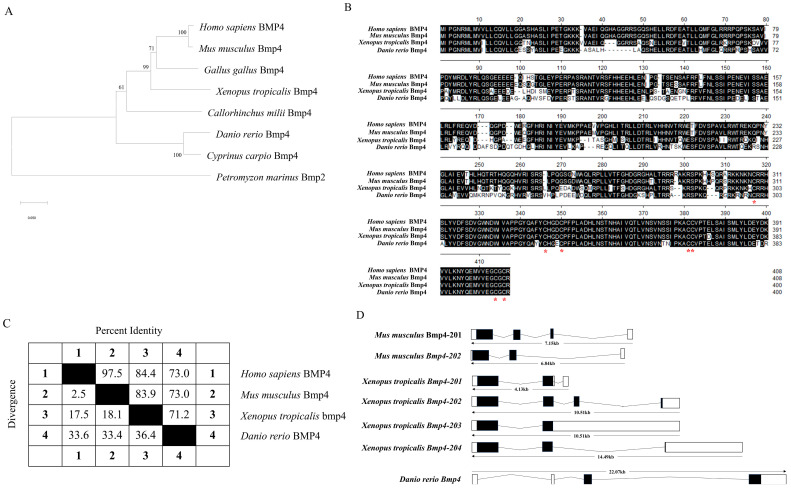
Bioinformatic analysis of *D. rerio bmp4*. (**A**) Phylogenetic tree of BMP4 proteins was constructed using MEGA (version 11) using the neighbor-joining method and sea lamprey Bmp2 was rooted as an outgroup. The reliability of each node was estimated by bootstrapping with 1000 replications. The numbers shown at each node indicate the bootstrap values (%). The bars represent the distance. (**B**) Multiple alignment of zebrafish Bmp4 and other known BMP4 amino acid sequences using the Clustal W program within the MegAlign of the DNASTAR software package (version 7.1). Shaded residues are the amino acids that match the consensus. The red asterisk represents the conserved cysteines. (**C**) Sequence similarity and sequence divergence of the BMP4 proteins in different species. Numbers in the table are calculated using the method of Clustal W in the software package DNASTAR (version 7.1). (**D**) Schematic representation of the gene organization comparisons of *BMP4* genes in different species. Exons are indicated with boxes and introns are represented as horizontal lines. GenBank accession numbers: *Homo sapiens* BMP4 (NP_001193.2); *Mus musculus* Bmp4 (NP_031580.2); *Gallus gallus* Bmp4 (NP_990568.4); *Xenopus tropicalis* Bmp4 (NP_001017034.2); *Cyprinus carpio* Bmp4 (XP_042630442.1); *Callorhinchus milii* Bmp4 (XP_007886339); and *Petromyzon marinus* Bmp2 (XP_032811632).

**Figure 2 ijms-24-14444-f002:**
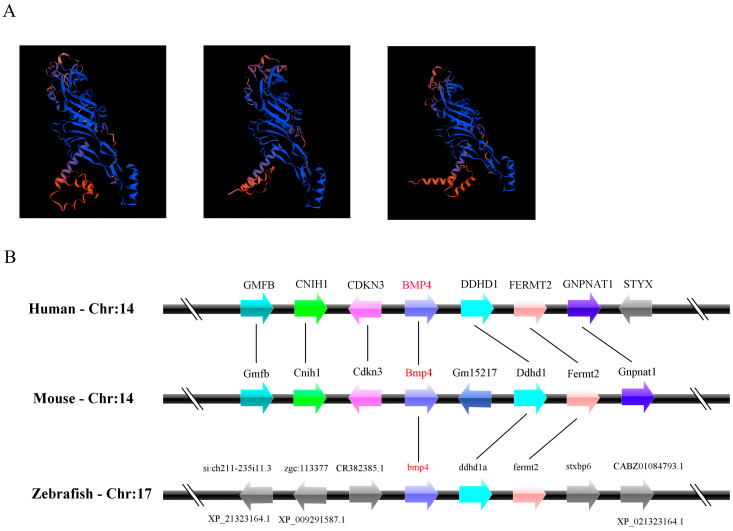
Comparison of the 3-D structure of BMP4 proteins and the loci of *BMP4* genes in different species. (**A**) The three-dimensional structures of *D. rerio* Bmp4, *H. sapiens* BMP4 and *M. musculus* Bmp4. This diagram was generated using SWISS-MODEL online software (https://swissmodel.expasy.org/, accessed on 2 July 2023). (**B**) Synteny map of the genomic segment with *BMP4* genes in different species. Genes are represented by boxes. Transcription direction is indicated by arrow.

**Figure 3 ijms-24-14444-f003:**
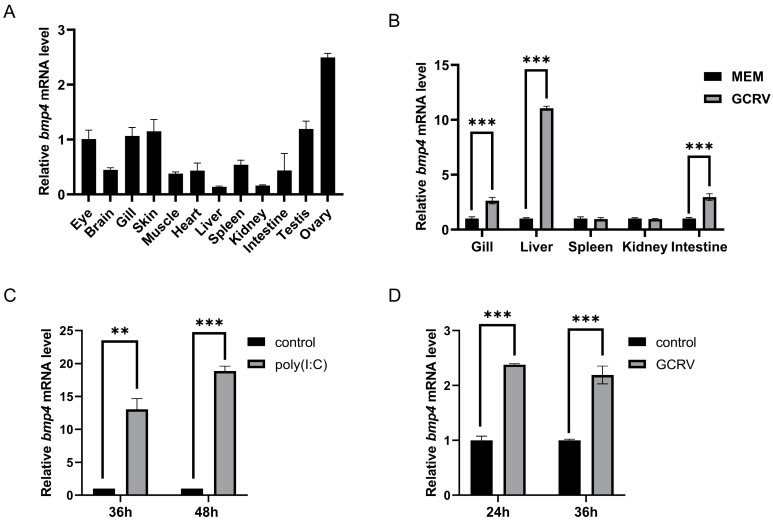
The *bmp4* expression was increased after virus or poly(I:C) challenge. (**A**) Distribution of *bmp4* mRNA in different tissues of adult zebrafish. (**B**) The expression of *bmp4* in the gill, liver, spleen, kidney and intestine from zebrafish challenged with GCRV. Zebrafish injected i.p. with MEM were used as the control. (**C**,**D**) Expression of *bmp4* mRNA in the ZFL cells challenged with poly(I:C) (**C**) or GCRV virus (**D**). ZFL cells challenged with PBS (**C**) or MEM (**D**) were used as the control. The expression of *actb1* served as an internal control for the qRT-PCR. Data were from three independent experiments and were analyzed through Student’s *t*-test (two-tailed). All data were presented as mean ± SD (** *p* < 0.01, *** *p* < 0.001).

**Figure 4 ijms-24-14444-f004:**
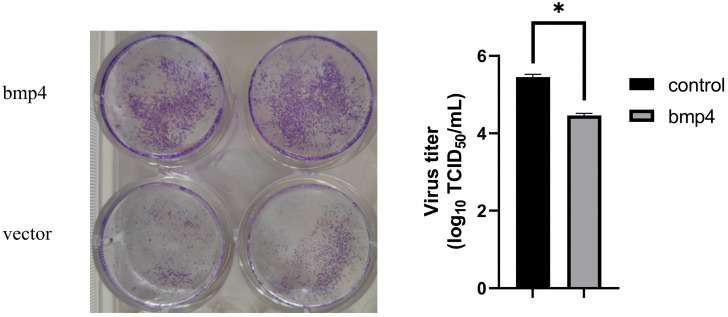
Antiviral function of Bmp4 in vitro. Overexpressing *bmp4* significantly decreased the viral titers in EPC cells. EPC cells were transfected with *bmp4* or empty vector, and 24 h later, the cells were infected with GCRV (10^6^ TCID_50_/mL). The cells and culture supernatants were collected 24 h post infection and the viral titers were determined through TCID_50_ assays. Data were analyzed using Student’s *t*-test (two-tailed) and are presented as mean ± SD (* *p* < 0.05).

**Figure 5 ijms-24-14444-f005:**
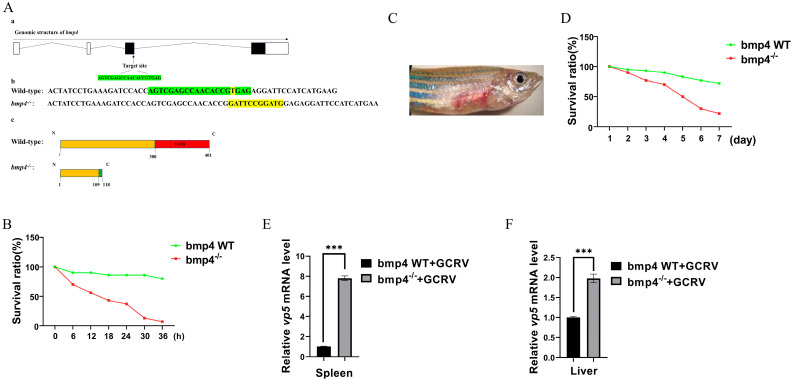
Antiviral function of Bmp4 in vivo. (**A**) The generation of *bmp4* mutation zebrafish using CRISPR/Cas9 technology. a. The knockout gene target is located in exon 3. The black box represents exons. The fold line represents introns. The white square represents un-translated regions. b and c. Compared with the wild type, 1 nucleotide was replaced by 11 nucleotides in exon 3 of bmp4 in the mutant, leading to early termination of translation to the 118th amino acid. (**B**) Kaplan-Meier ana-ysis of the overall survival of WT (*n* = 30) or *bmp4*^−/−^ zebrafish larvae (*n* = 30), which were infected by adding GCRV into the embryo culture medium (final 6 × 10^6^ TCID_50_/mL) and monitored every 6 h after infection. (**C**) The symptoms of adult zebrafish that were infected with the virus of GCRV. (**D**) Kaplan-Meier analysis of the overall survival of WT (*n* = 30) or *bmp4*^−/−^ zebrafish (*n* = 30) that were injected i.p. with 40 µL of GCRV (2.5 × 10^7^ TCID_50_/mL) and monitored every day after infection. (**E**,**F**) The expression of GCRV RNA in the liver and spleen from wild-type (WT) or *bmp4*^−/−^ zebrafish injected i.p. with 40 µL of GCRV (2.5 × 10^7^ TCID_50_/mL). The expression of zebrafish *actb1* was used as an internal control for the qRT-PCR. Data were from three independent experiments. Data were analyzed using Student’s *t*-test (two-tailed) and are presented as mean ± SD (*** *p* < 0.001).

**Figure 6 ijms-24-14444-f006:**
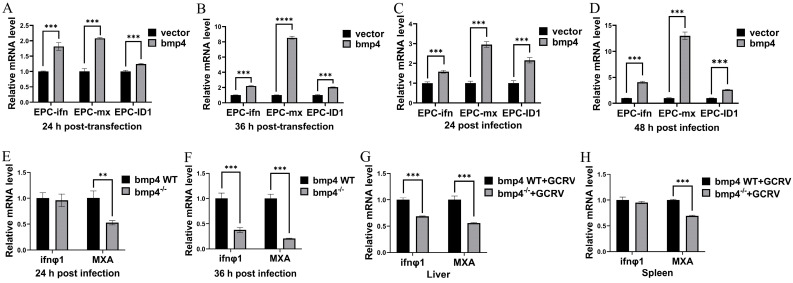
Bmp4 promotes the expression of antiviral genes. (**A**,**B**) Expression of EPC *ifn* and EPC *mx* mRNA after transfection with *bmp4* (2 μg) or empty vector (2 μg) in EPC cells. The cells were collected 24 h or 36 h post transfection. (**C**,**D**) Expression of EPC *ifn* and EPC *mx* mRNA after transfection with *bmp4* (2 μg) or empty vector (2 μg) in EPC cells for 24 h, followed by infection with GCRV for another 24 h or 48 h. (**E**,**F**) Expression of *ifnφ1* and *mxa* mRNA from WT or *bmp4^−/−^* zebrafish larvae challenged with GCRV for 24 h or 36 h. (**G**,**H**) Expression of *ifnφ1* and *mxa* mRNA in the liver and spleen from WT or *bmp4^−/−^* adult zebrafish challenged by injecting i.p. with 40 µL of GCRV (2.5 × 10^7^ TCID_50_/mL) for 36 h. The expression of zebrafish *actb1* or EPC *actin* was used as an internal control for the qRT-PCR. Data were from three independent experiments and were analyzed using Student’s *t*-test (two-tailed) for comparison of two groups. All data are presented as mean ± SD (** *p* < 0.01, *** *p* < 0.001, **** *p* < 0.0001).

**Figure 7 ijms-24-14444-f007:**
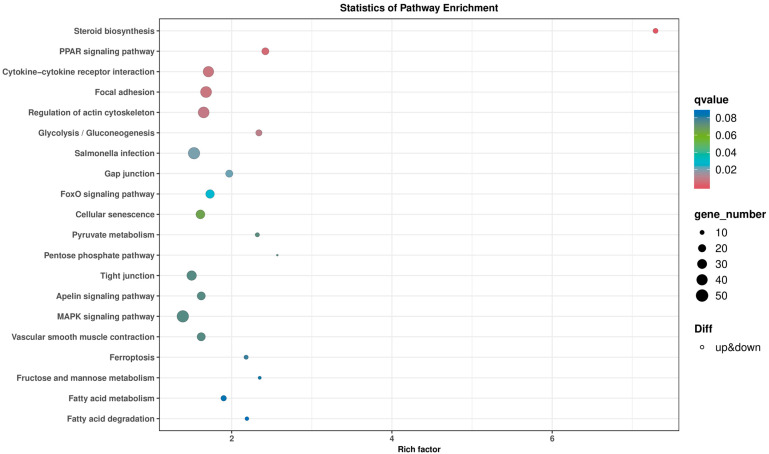
KEGG analysis of the pathways through enrichment of DEGs between *bmp4^−/−^* and WT liver tissue of zebrafish challenged by GCRV. Rich factor is the ratio of differentially expressed gene numbers annotated in these pathway terms to all gene numbers annotated in these pathway terms. The q-value means corrected *p*-value; q < 0.05 is significantly enriched.

**Figure 8 ijms-24-14444-f008:**
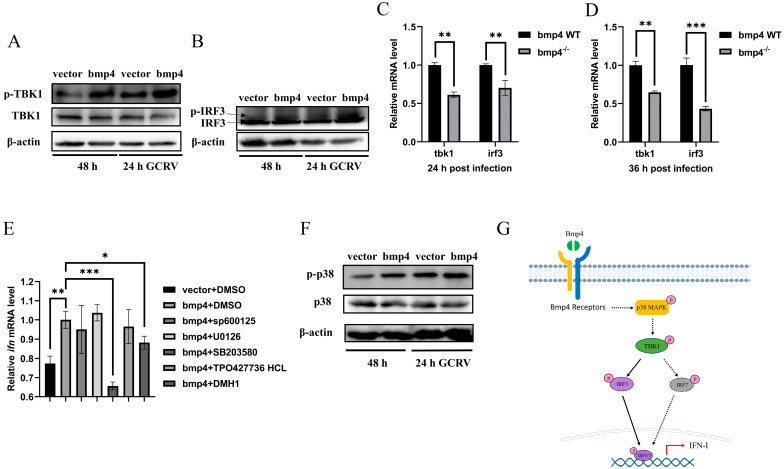
Bmp4 increases Tbk1-Irf3 antiviral signaling via p38 MAPK pathway. (**A**,**B**) Immunoblot analysis of phosphorylated (p-) Tbk1 (**A**) and Irf3 (**B**) was performed 24 h after transfection of 2 μg of bmp4 or empty vector in EPC cells, followed by infection with or without GCRV for 24 h. (**C**,**D**) Expression of *tbk1* and *irf3* mRNA from WT or *bmp4*^−/−^ zebrafish larvae challenged with GCRV. (**E**) Expression of EPC *ifn* mRNA 24 h after transfection of *bmp4* (2 μg) in EPC cells, followed by treatment with SB203580, SP600125, U0126, DMH1 and TP0427736 HCl for 24 h. (**F**) Representative Western blot analysis in p-p38 MAPK expression in *bmp4*-overexpressed EPC cells. (**G**) Schematic illustration of the regulation of Bmp4 in the antiviral immune response. The Bmp4 promotes phosphorylation of Tbk1 and Irf3 to induce the expression of *ifn* through p38 MAPK pathway. The expression of zebrafish *actb1* or EPC *actin* was used as an internal control for the qRT-PCR. Data were from three independent experiments and were analyzed through Student’s *t*-test (two-tailed) for comparison of two groups or one-way ANOVA followed by Games-Howell post hoc tests for comparison of multiple groups. All data are presented as mean ± SD (* *p* < 0.05, ** *p* < 0.01, *** *p* < 0.001).

**Table 1 ijms-24-14444-t001:** Sequences and applications of the primers used in this study.

Primer Name	Primer Sequence (5′—3′)	Application
EPC-bmp4-F	GTAGGCTGGAACGACTGGATTG	
EPC-bmp4-R	GCGTGATTGGTGGAGTTGAGA	
zebrafish-bmp4-F	CGCAGCCCTAAACAAAGAG	
zebrafish-bmp4-R	TGATTGGTGGAGTTGAGATGAT	
GCRV-vp5-F	CTCCCCGTGAGCGTATTT	
GCRV-vp5-R	GTTAGCAGCGGTAGTGACTTG	
EPC-ifn-F	ATAGACAACGCTAAGGTGGAGG	
EPC-ifn-R	TTCCGACGACTGCCTGTTC	
EPC-mx-F	GGGAGAAGGGATCAGTCATG	
EPC-mx-R	GGTTTAGTCAGAATACCGAGGG	
EPC-ID1-F	GATGTTGTCCGCTGCCTCT	qRT-PCR
EPC-ID1-R	CATGGTCATTTGCTCGTCC	
EPC-tbk1-F	TCAGAAGTTTGAGAACGGGAAGA	
EPC-tbk1-R	CGTAGACCACGATGCGGTGTAAG	
EPC-irf3-F	AACAAGAATGACACTGCGGA	
EPC-irf3-R	AACTCGGGAGGGACTTTCAT	
ZFL-ifnφ_1_-F	GTGGAGGACCAGGTGAAGTT	
ZFL-ifnφ_1_-R	GATTGACCCTTGCGTTGC	
ZFL-MXA-F	ATGGCTGGAGCAGGTGTT	
ZFL-MXA-R	TCTGTGGTGGCGATGTCA	
EPC-actin-F	TGTTCCAGCCATCCTTCTTG	
EPC-actin-R	TGATTTTCATTGTGCTGGGG	
ZFL-actin-F	GGTATTGTGATGGACTCTGGTGAT	
ZFL-actin-R	TCGGCTGTGGTGGTGAAG	
zebrafish-bmp4-F	ATGATTCCTGGTAATCGAATGCTG	cDNA cloning
zebrafish-bmp4-R	TTAGCGGCAGCCACACCC	

**Table 2 ijms-24-14444-t002:** Antibodies’ information.

Antibodies	Company	Product Code
Phospho-TBK1/NAK (Ser172) Rabbit antibody	Cell Signaling Technology	#5483T
TBK1/NAK Rabbit antibody	Cell Signaling Technology	#3504T
IRF3 Rabbit antibody	Bioss	#bs-2993R
Phospho-IRF3 (Ser386) Rabbit antibody	Bioss	#bsm-52170R
p38 MAPK Rabbit antibody	Bioss	#bs-0637R
Phospho-p38MAPK (Thr180 + Tyr182) Rabbit antibody	Bioss	#bs-2210R
β-Actin Rabbit antibody	Bioss	#bs-0061R
goat anti-rabbit IgG HRP secondary antibody	CWBIO	#CW0103S

## Data Availability

All relevant data are available from the authors upon request and the corresponding author will be responsible for replying to the request.

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
