# Peer review of "Bmp4 in Zebrafish Enhances Antiviral Innate Immunity through p38 MAPK (Mitogen-Activated Protein Kinases) Pathway"

_ijms, 2023, doi:10.3390/ijms241914444_

Round 1

Reviewer 1 Report

This work revolves around the characterization of bone morphogenetic protein 4 concerning its potential role against viral infections. It is a comprehensive and well-designed research work that facilitates the formulation of the stated conclusions. However, the manuscript in its current state requires revisions before it can be accepted. The main points for consideration are outlined as follows:

  1. Primarily, a significant amount of discussion elements have been noted within the Results section. It is strongly recommended that a thorough review be conducted to ensure that no discussion appears in the Results section. Furthermore, it has been identified that the discussion is excessively concise, a circumstance that might be attributed to the inclusion of some discussion aspects within the Results section. Hence, both sections should be rewritten.

2. Figure 1: This figure should be separated in several bigger figures since letters are too small to be read, especially for 1B.

3. Figure 3: The same as for figure 1.

4. Figure 4: Letters are also very small. I recommend to reorganize the whole figure in order to increase the size of the letters.

5. Line 32:  Unprecise expressions as “and so on” should be substituted by more precise ones.

6. Line 59: “Orthologues of this gene were also found in mammals, birds, amphibians, teleosts and cartilaginous fish”, where does the authors found these orthologues? There is no reference in the text.

7. Line 240: when authors says that BMP2, 4 and 7 were mutants, are they referring to those mutants as defective? This should be clarified.

Line 314: Authors say that FC>2 was taken into account. Does it mean that only up-regulated DEGs were used in the transcriptomic analysis? Why?

Line 368: “Student’s t-test”

Minor spelling issues.

Reviewer 2 Report

I authors Chen et al. investigated antiviral immunity using zebrafish model. As far as I am concerned, the manuscript was well structured and the results were presented clearly. I also appreciate the authors for writing the manuscript eloquently. But I suggest the authors to consider some points discussed below and revise the manuscript accordingly.

The authors need to provide full form of the abbreviations used in the abstract. For instance, GCRV. Also for HBV in line no.48

The authors have not mentioned poly(I:C) in the methodology but it was mentioned in the results and discussion. So please include about poly(I:C) in methodology.  

Other than the above comments, the manuscript was well written.

Reviewer 3 Report

The article by Lihui Chen and colleagues, titled "Bmp4 in Zebrafish Enhances Antiviral Innate Immunity 2 through p38 MAPK Pathway," is scientifically very interesting. The manuscript is cohesive throughout and well structured. However, I suggest that the authors deepen and improve the discussion section, perhaps by more accurately describing the p38 protein pathway. Also, more generally, more in-depth discussions , and the addition of others, more recent bibliographic data, could help to further improve the quality of the article. The authors could also improve the description of materials and methods. For example, they could create a summary table with details of the antibodies used for western blot, so as to improve understanding and reading of the results.
